# Review of Microwaves Techniques for Breast Cancer Detection

**DOI:** 10.3390/s20082390

**Published:** 2020-04-22

**Authors:** Maged A. Aldhaeebi, Khawla Alzoubi, Thamer S. Almoneef, Saeed M. Bamatraf, Hussein Attia, Omar M. Ramahi

**Affiliations:** 1Electrical and Computer Engineering, University of Waterloo, Waterloo, ON N2L3G1, Canada; maldhaee@uwaterloo.ca (M.A.A.); smbamatr@uwaterloo.ca (S.M.B.); oramahi@uwaterloo.ca (O.M.R.); 2Community College of Qatar, Doha 7344, Qatar; Khawla.Alzoubi@ccq.edu.qa; 3Electrical Engineering Department, College of Engineering, Prince Sattam Bin Abdulaziz University, Al-Kharj 11942, Saudi Arabia; T.almoneef@psau.edu.sa; 4Electrical Engineering Department, King Fahd University of Petroleum and Minerals (KFUPM), Dhahran 31261, Saudi Arabia

**Keywords:** microwaves, breast cancer imaging, dielectric properties of tissues, microwave imaging, image reconstruction

## Abstract

Conventional breast cancer detection techniques including X-ray mammography, magnetic resonance imaging, and ultrasound scanning suffer from shortcomings such as excessive cost, harmful radiation, and inconveniences to the patients. These challenges motivated researchers to investigate alternative methods including the use of microwaves. This article focuses on reviewing the background of microwave techniques for breast tumour detection. In particular, this study reviews the recent advancements in active microwave imaging, namely microwave tomography and radar-based techniques. The main objective of this paper is to provide researchers and physicians with an overview of the principles, techniques, and fundamental challenges associated with microwave imaging for breast cancer detection. Furthermore, this study aims to shed light on the fact that until today, there are very few commercially available and cost-effective microwave-based systems for breast cancer imaging or detection. This conclusion is not intended to imply the inefficacy of microwaves for breast cancer detection, but rather to encourage a healthy debate on why a commercially available system has yet to be made available despite almost 30 years of intensive research.

## 1. Introduction

Breast cancer is one of the most common types of cancer among women worldwide. In 2019, the American Cancer Society reported that more than 40,000 women died due to breast cancer and estimated that more than 260,000 new cases of invasive breast cancer will be diagnosed among women in the U.S. [1]. The same report also confirmed the deaths of 500 men due to breast cancer and estimated that nearly 2700 cases more will be diagnosed in male patients. The mortality rate is higher for men than women due to a lack of awareness, as they may not guess that a lump is actually breast cancer. Detecting breast cancer during its early stages of development is a fundamental factor for successful treatment [1]. The most common clinical imaging and detection modalities used for breast cancer detection are: X-ray mammography, magnetic resonance imaging (MRI), and ultrasound scanning [2].

Mammography is the only USFDA approved exam to be used for screening breast cancer in women with no prior symptoms. Mammography, however, has recently been subjected to immense scrutiny because of relatively high false negative and false positive results that can prove to be emotionally disruptive to the patient’s life [3]. Furthermore, women who use mammography as a screening test have a higher chance of developing cancer because of the ionizing radiation associated with X-rays [4,5]. Physical discomfort to women undergoing mammography is another drawback of this imaging technique [6]. Ultrasound screening is another technique of breast cancer detection where sound waves are transmitted through a transducer, which sends pulses into the breast and detects echoes from inside the breast; these echoes are used to form ultrasound images. However, ultrasound is not advantageous for breast imaging as it exhibits low resolution due to it not being able to distinguish between malignant and benign breast tumours [7,8]. MRI is highly sensitive in detecting invasive and small abnormalities compared with mammography and ultrasound techniques and can be used effectively for patients with dense breasts. Testing using MRI, however, is relatively expensive. Additionally, in MRI, inadequate breast positioning can cause unsuccessful detection [9].

The disadvantages and limitations of the current clinical detection techniques motivated researchers to investigate and develop new imaging techniques based on microwaves [10,11]. The concept of using microwaves for breast cancer detection received significant attention and extensive investigation from several research groups because of several advantages including low cost, having harmless radiation, and ease of use compared with current techniques, as shown in Table 1 [12,13]. In addition, the microwave imaging (MI) technique provides better sensitivity and the ability to detect small breast tumours because it is based on the electrical properties’ contrast between normal and tumours breast tissues [13].

The specific aim of this paper is to provide a thorough review of the microwave-based imaging techniques developed for breast cancer detection. The review focuses on the two types of MI, namely microwave tomography and radar-based techniques. A survey of the imaging algorithms, imaging methods, imaging system, and clinical prototype system is provided.

The other objective of this paper is to provide a chronological summary of research related to microwaves for breast cancer detection in order to highlight the challenges and limitations faced and to accentuate the incremental improvements accomplished over the years. Specifically, the following chronological summaries are provided: (1) a chronological summary of studies and measurements of the dielectric properties of human female breast tissues; (2) a chronological summary of research related to microwave tomographic imaging methods and systems; (3) a chronological summary of research related to radar-based methods and systems. Compared to other similar works, the presented review paper is a unique comprehensive chronological summary that covers all aspects of MI for breast cancer detection citing and discussing approximately 300 references. Additionally, this work addresses the challenges (i.e., sensitivity, cost, harmful radiation, and inconveniences to the patients) that the current detection systems are facing with a detailed discussion of various software algorithms, antenna and sensors types, and operating frequencies.

## 2. Microwave Dielectric Properties of Breast Tissues

In recent years, microwaves have emerged as a potential technique for breast cancer detection that avoids the discomfort, high-risk, and costs associated with X-rays and MRI. The fundamental premise of microwave detection is the sharp contrast in the dielectric (electrical) properties of cancerous tumours in comparison to healthy tissues. To avoid all the limitations mentioned in the current techniques, researchers have shifted their attention to an alternative modality based on microwaves. MI is a technique used to detect hidden or embedded objects inside a medium using microwaves. An MI system is based on two main parts: (i) hardware (antenna) responsible for illuminating and collecting microwave signals reflected from the illuminated object under test such as breast tissues; (ii) software or post-processing algorithms that are responsible for reconstructing an image of the target object as shown in Figure 1 [8].

Knowledge of the dielectric properties of breast tissues is essential for the understanding of the interactions between the electromagnetic fields (i.e., microwaves) and the breast tissue. The dielectric properties of breast tissues are represented by a complex permittivity where the real part represents the ability of the material to store microwave energy, whereas the imaginary part, or the loss factor, indicates the ability of the material to absorb microwave energy [14]. Microwave-based detection modalities are mainly based on the observation of a high variation of the dielectric properties between normal and malignant breast tissues [12,13]. Microwaves reflected off or scattered from breast tissues are expected then to result in an image showing a contrast between malignant and healthy tissues.

Several research groups studied the dielectric properties of normal and malignant breast tissues in the microwave region. These studies established that the malignant breast tissues have significant differences in their dielectric properties in comparison to nearby healthy breast tissues. Table 2 gives a chronological summary of several published studies of the dielectric properties’ contrast between normal and malignant human breast tissues [13,15,16,17,18,19,20,21,22,23]. The table is subdivided into five categories: the year of study, the author names, the operating frequency, the probe used in measurements, and the number of samples. The accuracy of the difference between the normal and tumourous tissues of each reported study is based on many factors such as the frequency range for the fitting model, tissue type and size, number of samples used for the study, sensing depth and volume, time from excision to measurement of tissues in the study, type of the fitting data model used, and the environment of testing (i.e., in vivo or ex vivo states) [19,20,24,25,26,27].

Several studies have investigated the contrast between the dielectric properties of normal and malignant breast tissues [13,17]. The studies demonstrated by Chaudhary et al. 1984, Surowiec et al. 1988, and Joines et al. 1994 had some limitations for estimating the differences between the dielectric properties of normal and malignant breast tissues. The reasons for such limitations are the employment of a small number of samples (patients), the use of a small range of frequencies not exceeding 3.2 GHz, and ignoring the study of the dielectric properties’ differences between the three types of normal, benign, and malignant breast tissues [19]. On the other hand, the study introduced by the University of Wisconsin and the University of Calgary reported the dielectric properties of normal, benign, and malignant breast tissues within an ultra-wideband frequency range. However, this study had some limitations as they roughly used a sample volume depth on the order of the magnitude. Further, they used a Cole–Cole model for data fitting to plot the data in a linear fashion, which instead should have been plotted in a logarithmic fashion. This can have a significant impact on the measured data, apart from the effects of sample temperatures, patients’ age, and time taken between the measurements and excision of the samples [20,24,30]. The study reported in [20] introduced the differences in dielectric properties between in vivo and ex vivo states of breast tissues. However, this study considered a small sample of six women. The studies reported in [21,22,23,28] used larger samples and a wide range of frequencies for measuring the differences between the dielectric properties of normal and tumourous tissues. In summary, in spite of the many published studies on the contrast of the dielectric properties between normal and tumourous breast tissues, many challenges still exist. These challenges include the inaccurate estimation of the in vivo contrast, uncertainties in measurement methods, and the results not being in agreement with other standard detection methods.

With the advancement of the nanotechnology field and nanomaterials, several studies [31,32,33] proposed and investigated the possibility of improving the available cancer detection methods (magnetic resonance imaging (MRI), magneto-acoustic tomography (MAT), computed tomography (CT), and near-infrared (NIR) imaging) by combining the detection method with magnetic nanoparticle materials that are biocompatible such as iron oxide nanoparticles. In microwave detection for breast cancer, some recent studies explored the advantages of using magnetic nanoparticles to improve the contrast between the dielectric properties of normal, benign, and malignant breast tissues, as well as the in vivo and ex vivo states of breast tissues [31,32,33,34].

## 3. Microwave Imaging for Breast Cancer Detection

Microwave-based detection techniques offer several advantages over other detection methods such as being inexpensive, non-invasive, non-ionizing, and a comfortable form of treatment [13,20]. In addition, MI techniques provide higher sensitivity and the ability to detect small breast tumours as these techniques are based on the contrast of electrical properties between normal and tumourous breast tissues [13,20]. Such detection techniques are based on the hypothesis that the electrical properties, namely the permittivity and conductivity of malignant breast tissues, differ from those of healthy breast tissues within the microwave band [35,36]. In literature, there are three modalities that have been explored for microwave-based breast detection. As seen in Figure 2, these methods can be categorized as passive, hybrid, and active methods [35,36,37,38,39].

In the passive approach, a radiometry device is utilized to measure the differences in temperature between healthy and malignant breast tissues. Tumourous tissues exhibit a higher temperature compared to the surrounding normal tissues due to differences in electrical properties [38,39]. Hybrid techniques, termed thermoacoustic methods, employ microwave sensors and ultrasonic transducers to detect the presence of a tumour. In the hybrid techniques, the breast tissues are illuminated by microwaves, which are then absorbed by them. Malignant tissues significantly absorb more energy than normal tissues and radiate stronger acoustic waves due to their higher electrical conductivity than that of the surrounding normal tissues. An ultrasound transducer located close to the breast then measures the reflected acoustic signals, which are then processed to reconstruct an image [40,41]. Active MI employs microwave signals to illuminate the breast tissues. The variation in the dielectric properties between malignant and normal breast tissues is then measured within specific microwave frequency range [42]. Such a technique estimates the back-scattered signals from the breast tissues to reconstruct an image that indicates the presence of a tumour by virtue of stronger reflected signals than those of healthy tissues [42,43,44].

Techniques for breast cancer imaging using active microwave methods are extensive, including tomography and radar-based techniques [44,45]. The tomography approach is usually performed iteratively and can be represented by a nonlinear inverse problem that requires significant computational resources for producing the dielectric properties of the breast. Imaging of the breast tissues can be constructed from the recovered microwave data file through inverse scattering algorithms that estimate the constitutive parameters of the breast tissues by analysing the absorbed and reflected microwave signals [46,47]. The radar-based approaches utilize an external microwave source to illuminate the breast tissues with ultra-wideband (UWB) signals. The back-scattered signals from the breast are used for detecting breast tumours [48,49,50].

### 3.1. Microwave Tomographic Imaging

Microwave tomography (MT) applies microwave signals that illuminate the breast tissue to reconstruct the permittivity and conductivity of the breast. This approach aims at the complete reconstruction profile of the dielectric properties of the imaged object. It generally leads to an ill-posed nonlinear inverse scattering problem, which reconstructs the image by estimating the scattered fields and compares it with the signals obtained from a normal breast [46]. The MT method has been investigated in theoretical and experimental studies for breast cancer detection by several research groups [46,51,52,53,54]. MT uses three steps to reconstruct an image of the breast tumours: collecting microwave tomogram data, analysing the data, and finally, displaying the images [47,55]. The acquisition of the microwave data is performed by exposing the breast tissue to a microwave signal and then producing the microwave tomogram data file. The microwave tomogram data analysis is performed using inverse scattering algorithms that calculate the dielectric properties of the normal and tumourous breast tissues from the microwave tomogram data files. The data visualization part is achieved by utilizing 2D or 3D tomogram images that show the presence, location, and size of the tumour [37,54]. The MT system for breast cancer detection combines both a hardware part that includes microwave sensors and probes and a software part that includes image reconstruction approaches and algorithms.

Reconstruction methods are an essential part of MT to reconstruct an image of breast tumours. In the literature, there are two different methods that have been used for solving the nonlinear inverse MT problems for breast imaging, namely the gradient-based local and global methods [56,57,58,59]. In both techniques, the MT problem is formulated as a cost function, which is solved through an iterative process.

In the local gradient-based approach, the nonlinear inverse problems are solved by using several data-processing algorithms developed for breast cancer detection such as the conjugate gradient least squares (CGLS) algorithm, least squares QR (LSQR) algorithm, and Landweber algorithm [58,59,60]. The advantage of using the gradient-based algorithms is their low processing time due to the smaller number of iterations [37,61]. The global method is able to reach the global minimum of a cost function through randomization, which searches for several local minima in the entire solution space and is not limited to a search only in the gradient direction. The global methods are based on evolutionary algorithms such as the genetic algorithm (GA) and particle-swarm optimization (PSO) algorithm [37,61]. There are several image reconstruction algorithms that have been developed and used in microwave tomography imaging approaches to solve complex ill-posed nonlinear problems for breast cancer detection [60,62,63,64,65,66].

In this sub-section, some of the most famous tomography MI systems in the literature are reviewed. We consider some factors that have an impact on each system such as the operating frequency, the type of antenna used, the number of channel data acquisition systems, the array type, the scanning time of examination, the coupling medium, the number of clinical trials, the type of algorithms used for reconstructing a 2D or 3D tomogram image, and most importantly, the size of the smallest detected tumour. These features are especially essential for showcasing the performance, accuracy, complexity, and simplicity of each reviewed system.

The first clinical prototype for the near-field tomography MI system for breast cancer detection was reported in [44]. In this work, a 32 channel data acquisition subsystem was developed, which incorporated a heterodyning modulation scheme, a 200 kHz A/D board, and a signal phase and amplitude extraction. The system consisted of a cylindrical array of a transceiving monopole antenna array (16 elements) in the 300–1000 MHz frequency range and an HP85070B dielectric probe kit in conjunction with an HP 8753C network analyser [44]. All components and modules were consolidated into a single cart for the purpose of portability. The fixed synthetic array of antennas was placed on an examination table and connected. Saline was used as the coupling medium between the breast and the antenna array. The architecture required the patient to lie down flat on a bed while placing the breast into the MI bath under the bed, as shown in Figure 3 [44]. The microwave clinical exam was conducted upon five women, and the scan time sessions taken were between 10 and 15 min per breast. Figure 4a,b shows the resultant images of the recovered 900 MHz relative permittivity and conductivity images for both left and right breasts, respectively, of Patient 1.

The same group also developed the first 3D prototype tomography MI system based on the 3D finite element modelling (FEM) method for microwave tomography [67]. The new developed system overcame some limitations in both the hardware and software aspects of the previous system. The system consisted of an array of 16 monopole antennas. Hydraulic seals in the tank base were designed to allow free vertical antenna motion over the full tank span, ensuring proper data acquisition from the chest wall to the nipple region for all breast sizes, as shown in Figure 5. The system was capable of measuring signals down to levels compatible with sub-centimetre image resolution, while keeping the exam time well under two minutes and the duration of the processing time less than 20 min. The reported ultra-fast system had unique synergy between the hardware and software as both were optimized to achieve a near real-time imaging with no compromise in accuracy. The antennas were divided into two interleaved sets of eight antennas controlled by separate motors for their vertical positioning, as illustrated in Figure 5. The results from clinical trials demonstrated that a tumour with a size of 1.5 cm × 1.2 cm × 1 cm was detected in the right breast of a 55-year-old patient, as shown in Figure 6 [67].

Son et al. developed a preclinical prototype MT system for breast cancer detection, as shown in Figure 7a [68]. In this system, a 16 element circular array was placed into an imaging bath having a matching liquid. Each antenna was used for signal transmission and reception over a frequency band ranging from 500 MHz to 3000 MHz. A matching liquid was used to fill the MI bath to reduce reflections from the breast surface, as shown in Figure 7b. The total number of transmit-receive data points was 240 (measurements were performed for each frequency at each imaging plane). The data acquisition unit used a programmable switch matrix, as shown in Figure 8 [68].

The proposed 2D tomography system in [68] is convenient for trials on patients and phantoms. The reported prototype system was used to detect a 10 mm tumour inside a phantom. Figure 9 shows the reconstructed images for the breast phantom with a cylindrical 10 mm tumour at 1700 MHz. Figure 9 shows the image of a breast phantom without a tumour, with a tumour positioned at the centre of the breast, and with a tumour positioned 30.0 mm right of the centre.

Pagliari et al. [69] developed the first prototype for a tomography MI system based on low-cost off-the-shelf microwave components, custom-made antennas, and a small form-factor processing system with an embedded field-programmable gate array (FPGA) for accelerating the execution of the imaging algorithm (see Figure 10). This prototype system was fast, accurate, and relatively low in price mainly because it was configured by an FPGA that was capable of executing the imaging algorithm 20 times faster than a multi-core CPU. Furthermore, this prototype system [69] was capable of detecting a tumour with an accuracy similar to the one that could be achieved using expensive equipment such as a vector network analyser (VNA). For image construction, a linear inversion method (I-MUSIC) was used instead of a non-linear inversion algorithm. The system operated over a relatively small bandwidth of 200 MHz between 1.4 and 1.6 GHz (see Figure 11). The relatively low cost for this system was maintained due to the availability of low-cost off-the-shelf components in the working frequency range of the system. Figure 12 depicts that a 40%–60% glycerine and water mixture was used to fill a 20 mm diameter plastic dielectric cylinder tank, which was intended to mimic the dielectric properties of a 2D breast phantom. A 12 mm diameter metallic cylinder was intended to mimic a tumour (see Figure 12). Figure 13 shows that the experimental results demonstrating the successful detection of a tumour having a size of 12 mm in diameter [69].

Jeon et al. introduced a clinical trial prototype MT system for detecting breast tumours in [70]. This clinical trial prototype system was designed to work in the range of 3 to 6 GHz with a fast forward solver (FFS) algorithm. Figure 14 shows this MT clinical trial system at a hospital in Seoul. The clinical trial prototype system was used for investigating 15 women ranging in age from 40 to 68 years [70]. Clinical trial results illustrated that a tumour with a size of 25 mm could be detected in the left breast (see Figure 15) [70].

### 3.2. Radar-Based Microwave Imaging

Radar-based techniques employ an external microwave source to illuminate breast tissue with ultra-wideband (UWB) signals. Back-dispersed signals from the breast are used to detect breast tumours. Bridges et al. proposed the first radar-based MI system in 1997 [71]. It avoids using nonlinear inverse scattering algorithms for recovering the complete profile of the breast’s dielectric properties to construct an image of the breast [72,73]. Bridges’ method utilized an ultra-wide microwave signal to illuminate the breast tissues using an antenna array placed at different locations surrounding the breast [73,74]. The back-scattered microwave signals were then received by the same antennas and processed by radar-based MI methods to detect the tumour’s location and size. These back-scattered signals indicated the presence of the tumour because of the contrast in the dielectric properties between the normal and tumourous tissue at microwave frequencies. Radar-based MI techniques involve radar-based MI methods and radar-based MI beam-forming algorithms [75]. Radar-based MI methods are based on different radar configurations that employ elements of an antenna array that transmit or receive (separately or simultaneously) microwave signals into or from the breast. The beam-forming algorithms are used to reconstruct images from the reflected signals.

Radar-based MI techniques can be classified into five approaches: confocal microwave (CM), tissue sensing adaptive radar (TSAR), MI via space-time (MIST), multi-static adaptive (MSA), and time-domain data adaptive (TDDA). The beam-forming algorithms can be grouped into several imaging algorithms for breast tumour detection including delay and sum logarithm (DAS), time reversal (TR) algorithms, artefact removal algorithms (AR), time-of-arrival algorithm (TOA), skin subtraction algorithm (SSA), inverse scattering algorithm (ISA), Gauss–Newton iterative algorithm (GNIA), matched-filtering algorithm (MFA), generalized likelihood ratio test (GLRT), constrained robust capon algorithm (CRC), inverse fast Fourier transform algorithm (IFFT), local discriminant basis (LDB), forward-backward time-stepping (FBTS), time-shift-and-add algorithm (TSAA), linear constraint minimum variance (LCMV), genetic algorithm (GA), filtered delay-and-sum (FDAS), skin reflection removal (SRR), double constrained robust Capon beamforming (DCRCB), norm constrained Capon beamforming (NCCB), empirical mode decomposition (EMD), modified delay-and-sum (MDAS), delay-multiply-and-sum (DMAS), and robust and artefact resistant (RAR).

This section reviews these methods and algorithms that were specifically used for breast cancer detection. We summarize and tabulate these methods and algorithms in Table A1 in Appendix A. These tables classify the methods and algorithms into seven categories: the year of the study, the author names, the operating frequency, the number of antennas employed, the tumour size, the radar-based MI techniques, and the image reconstruction algorithms.

Klemm et al. designed and implemented a UWB frequency-domain radar-based MI system [76]. The system consisted of an aperture array of UWB antennas that were positioned on a section of the hemisphere that conformed to the curved breast shape, as shown in Figure 16a. The UWB array was positioned with the breast comfortably resting on the spherical shell, as shown in Figure 16b. The signals were captured by a data acquisition module and transferred to a computer [76]. Clinical imaging results showed that the system could detect an 8 mm tumour, as shown in Figure 17.

Zhurbenko et al. reported a radar-based frequency-domain system [77]. The architecture of the system consisted of 32 monopole antennas, a measurement unit, a data acquisition system, and a computer. The antennas were used to measure the amplitude and phase of the scattered fields in the 3D imaging domain using electronic scanning in the frequency range from 0.3 to 3 GHz [77]. Fear et al. developed a 3D UWB tissue sensing adaptive radar (TSAR) MI prototype system for breast cancer detection [78]. This system consisted of a patient interface employing a padded table with a hole through which the breast dropped into a cylindrical container, as shown in Figure 18 and Figure 19. The mono-static approach was used to collect data, and a filtering circuit was used to reduce reflections from the skin tissues. The system was tested on eight volunteers ranging in age from 32 to 47 years [78]. The system was able to detect breast tumours of 5 mm in diameter, as depicted in Figure 20.

Kikkawa et al. developed a radar-based time-domain MI system using impulse radio ultra-wide-band complementary metal oxide semiconductor (IR-UWB-CMOS) integrated circuits [79]. The CMOS circuits consisted of a transmitter circuit, an UWB low noise amplifier (LNA), an equivalent time sampling circuit, a Gaussian mono-cycle pulse (GMP) generator, and an analogue-to-digital converter. The UWB antennas were used to transmit digitally generated GMP signals. A confocal imaging algorithm was used to transform the received signals to pixels, as shown in Figure 21 [79].

Porter et al. introduced an experimental multi-static radar-based time-domain MI system for breast cancer detection, as shown in Figure 22 [80]. The system consisted of a radome, which was the interface between the system and the breast that was positioned inside it, a 16 element antenna array, a commercial pulse generator, an oscilloscope, a switch matrix, and a computer (see Figure 23). The system used time-domain measurements. A proof-of-concept was demonstrated by imaging breast tumours in realistically-shaped breast phantoms. The system detected a spherical tumour with a radius of less than 1 cm inserted in a realistically-shaped breast phantom, as shown in Figure 24 [80].

Kwon et al. proposed a complementary metal-oxide semiconductor (CMOS) time-domain MI system for breast cancer imaging [81]. The proposed MI system consisted of 16 UWB transceivers that surrounded a 3D hemispherical phantom and a master controller for data collection. The data were processed using the multi-static delay and sum (DAS) algorithm to reconstruct an image of a 3 mm tumour inserted inside a 3D hemispherical breast phantom model [81]. Santorelli et al. designed and implemented a 3D radar-based time-domain MI system [82]. The system was compact, cost-effective, and wearable. The system consisted of a flexible circuit with 16 UWB embedded antennas, two micro-controllers, a shaping circuitry, a recording oscilloscope, an impulse generator, and a 16 × 2 switching matrix network implemented using solid-state technology (see Figure 25). The switching matrix allowed the antennas to operate independently as either transmitters or receivers. The system performance was clinically tested showcasing its capability for detecting a 1 cm tumour inserted inside a breast phantom (see Figure 26) [82].

Porter et al. developed a clinical prototype of a wearable radar-based time-domain MI system with a wearable patient interface for microwave breast tumour detection [83]. The wearable system consisted of 16 flexible monopole antennas distributed asymmetrically around a bra surface. The developed system had two good features compared with other systems including portability and compactness as the system did not require an examination table. The antennas did not have direct contact with the skin, which in turn eliminated the need for an immersion medium. Additionally, due to the close fit with the fabric that hosted the antenna array, the position of the breast relative to the array was known, which eliminated errors in the images arising from using the immersion liquid. A schematic of the system measurement setup is provided in Figure 27a, where a pulse is generated and amplified then fed into a switching network, which selects the transmitting and receiving antennas. The received signal was recorded by a picoscope. The wearable prototype was used for clinical testing on a healthy volunteer over a 28 day period. The results showed that the quality of reconstructed breast images was improved (see Figure 28) [83].

Li et al. presented a cost-sensitive ensemble classification technique for microwave breast tumour detection [84]. They used two different sets of data for the classifications, one based on clinical experiments, which were collected every month in a clinical trial from 12 healthy volunteers over an eight-month period, and the other was based on heterogeneous breast phantoms. Simulated tumour responses were artificially added to existing data scans of healthy volunteers in order to assess the efficacy of the proposed classification techniques. The system consisted of 16 element antenna sensors to gather the illuminated and reflected signals from the breast tissues. The classification techniques were based on three fusion strategies to perform classification using cost-sensitive support vector machines. The presented classification techniques had the ability to distinguish between normal and malignant breast tissues [84]. A summary of some clinical radar-based MI studies is shown in Table 3. This table compares various clinical studies in terms of seven aspects: the scan time, maximum number of patients, the coupling medium, the examination table, the antenna array type, the acquisition and imaging, and the position.

Some other studies as shown in Table A2 focussed on the algorithms and methodologies employed for breast cancer detection, rather than on the hardware apparatus. This table summarizes the most recent innovative and effective methodologies for breast cancer imaging [85,86,87,88,89,90,91,92,93,94,95,96,97,98,99,100,101]. Those methodologies aimed at achieving multi-resolution images. Furthermore, some of these methodologies adopted magnetic nanoparticles [87,88] and compressive sensing tools [97] to enhance the detection process, by exploiting linear approximations of the inverse scattering problem or iteratively solving a set of linear problems [99,100,101].

## 4. Conclusions

Breast cancer remains the second-leading cause of death among women worldwide. Microwave imaging, which is noninvasive, non-ionizing, inexpensive, and harmless to humans, offers a promising alternative technique for breast tumour detection and for breast imaging. Over the past 30 years, significant progress has been made towards producing microwave imaging techniques for breast cancer detection. This paper presented an exhaustive summary of research in microwaves as it relates to breast cancer imaging and detection. Specifically, the review showed several aspects of the imaging modalities including imaging algorithms, imaging methods, sensors used, and the use of artificial intelligence. All the studies conducted so far gave a strong indication that microwave imaging has a high potential for both imaging of the human female breast and cancer detection. However, research showed that some significant challenges have been reported regarding practical implementations. The most important of these challenges faced were the need for high sensitivity sensors and insufficient image resolution.

Additionally, most commonly developed systems exhibited several other challenges and limitations that included: using multiple antennas with multiple feeds that led to a complex system, using a matching liquid between the antenna and the skin that led to a low signal-to-noise ratio (SNR) and a decrease in the performance of the system, using complex computations for reconstructing an image of the breast tissues and complex coding for image processing, low resolution and accuracy arising from the mutual coupling between multiple antennas, and high cost because of the use of multiple antennas with multiple feeds that required expensive equipment such as oscilloscopes, signal generators, and VNAs.

Despite numerous studies and significant funding specifically related to the use of microwaves for breast cancer detection, and after more than 30 years of active research on the subject, a commercial and cost-effective microwave-based system is still not available. It is difficult to provide a satisfying answer to this crucial question. Nonetheless, it is perhaps timely to revisit the fundamental modalities that have been employed, mainly the fundamentals behind breast image reconstruction, which are based on the mathematically ill-posed inverse problem and radar-based techniques. Both techniques were initially developed for completely different applications, but have been proposed and adopted for breast imaging. Additionally, both techniques were developed for much simpler engineering problems unlike the problem of imaging highly complex and highly dispersive human tissues or the detection of anomalies within such tissues. In fact, both techniques are fundamentally centred on far-field assumptions. It is hoped that these conclusions will encourage a healthy debate on why a commercial cost-effective microwave-based detection system has yet to be made available despite almost 30 years of intensive research. 

## Figures and Tables

**Figure 1 sensors-20-02390-f001:**
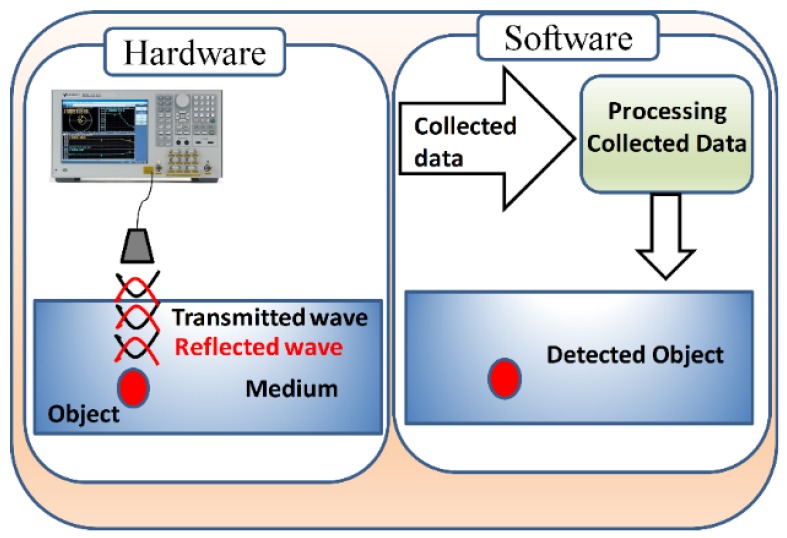
A schematic diagram showing the components of the microwave imaging (MI) system.

**Figure 2 sensors-20-02390-f002:**
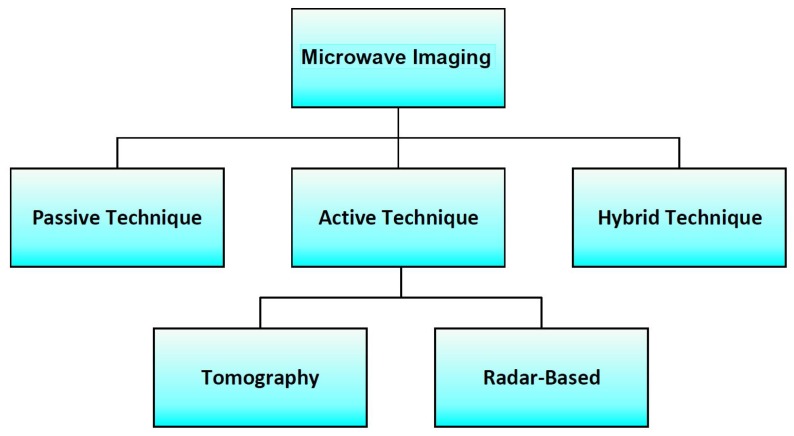
Block diagram showing the different modalities that have been explored for microwave-based breast cancer detection.

**Figure 3 sensors-20-02390-f003:**
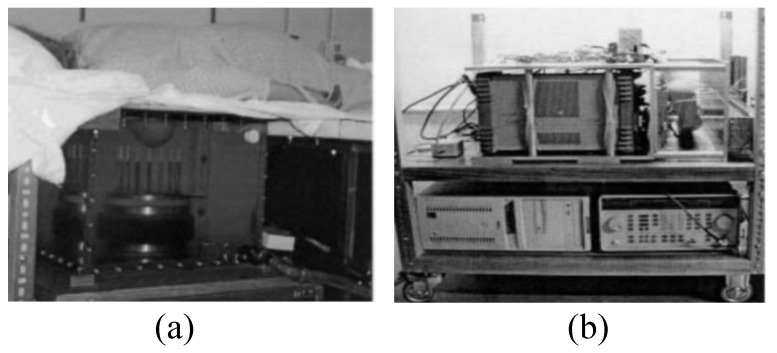
Prototype of the imaging system developed by Meaney et al. [44]. (**a**) View of the examination table. (**b**) View of the data acquisition system including the microwave source. Reprinted with permission from [44].

**Figure 4 sensors-20-02390-f004:**
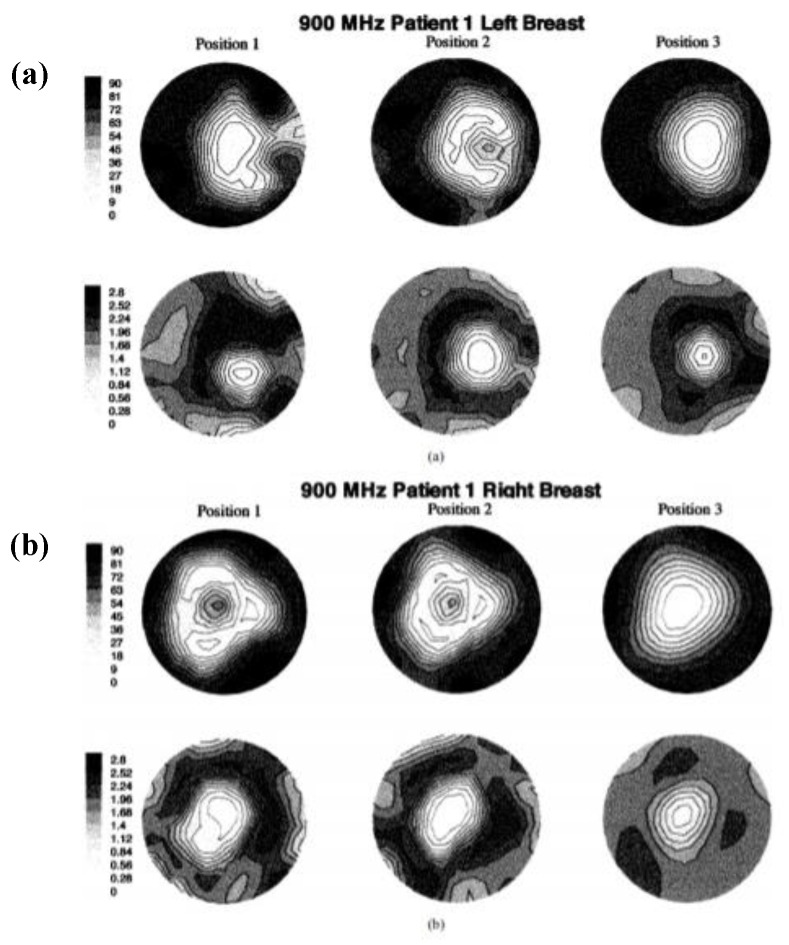
Recovered 900 MHz relative permittivity and conductivity images from Patient 1: (**a**) left breast and (**b**) right breast. Reprinted with permission from [44].

**Figure 5 sensors-20-02390-f005:**
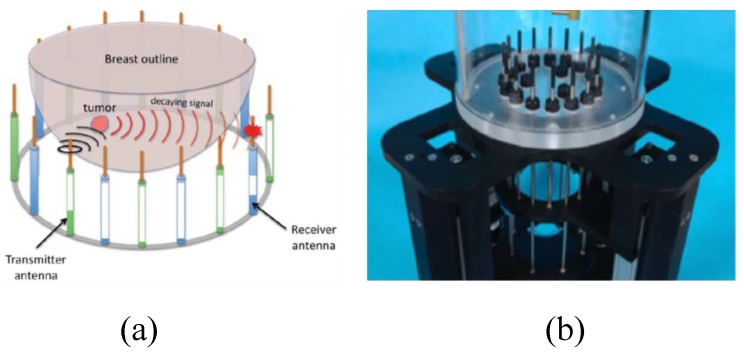
A prototype system for breast cancer testing developed by Grzegorczyk et al. [67]. (**a**) View of a 3D schematic representing the testing setup. (**b**) Photograph showing the system hardware. Reprinted with permission from [67].

**Figure 6 sensors-20-02390-f006:**
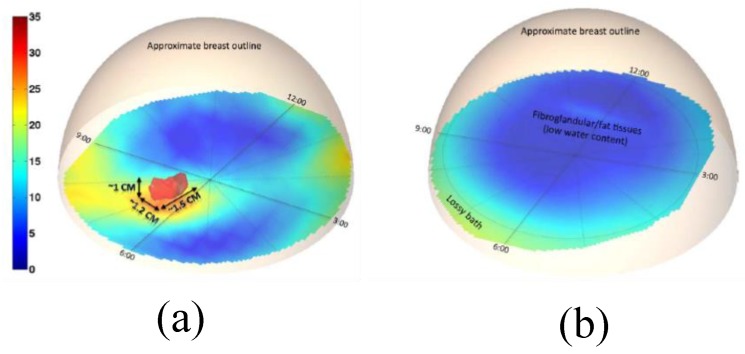
3D microwave tomographic permittivity images at 1300 MHz obtained from the system developed by Grzegorczyk et al. [67]. (**a**) The 3D structure of a tumour present in the right breast. (**b**) The 3D structure of a tumour-free left breast. Reprinted with permission from [67].

**Figure 7 sensors-20-02390-f007:**
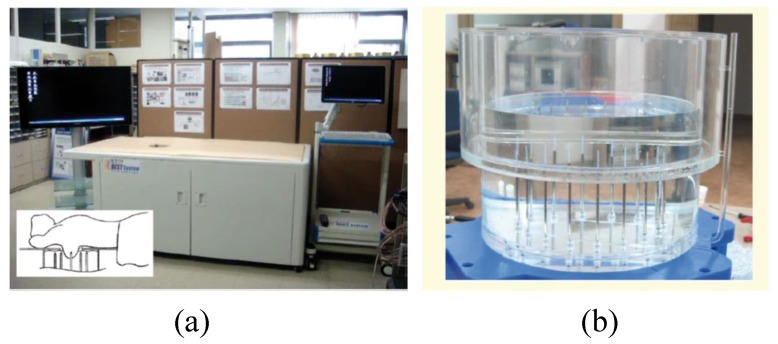
A prototype imaging system developed by Son et al. [68]. (**a**) View of the entire system. (**b**) View of the breast holder showing the antenna array in the liquid bath. Reprinted with permission from [68].

**Figure 8 sensors-20-02390-f008:**
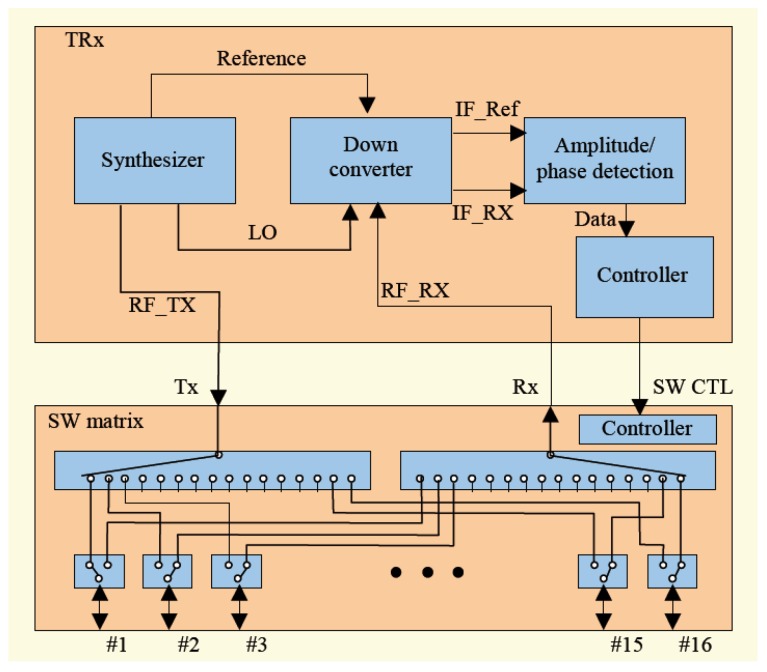
Schematic of the system developed by Son et al. [68]. Reprinted with permission from [68].

**Figure 9 sensors-20-02390-f009:**
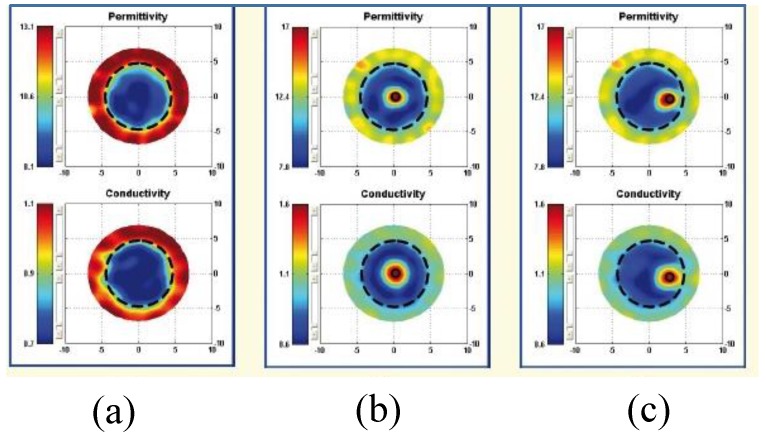
Reconstructed images at 1700 MHz for a breast phantom with a cylindrical 10 mm tumour. (**a**) Image of the breast without a tumour. (**b**) Image of the breast with a tumour positioned at the centre of the breast. (**c**) Image of the breast with a tumour positioned 30 mm right of centre. Marked big circles show the boundaries of the breast [68].

**Figure 10 sensors-20-02390-f010:**
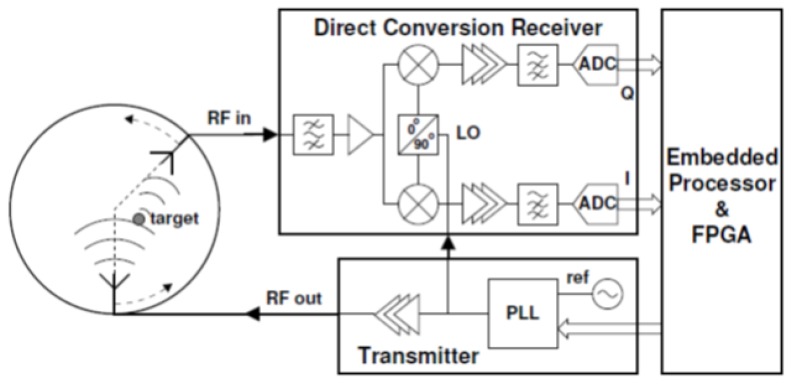
A schematic of the system developed by Pagliari et al. [69]. Reprinted with permission from [69].

**Figure 11 sensors-20-02390-f011:**
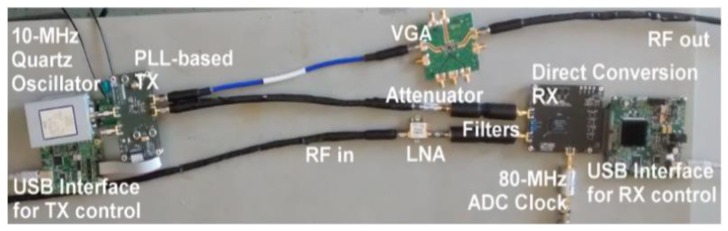
Photograph of the circuit used in the system developed by Pagliari et al. [69]. Reprinted with permission from [69]. LNA, low noise amplifier.

**Figure 12 sensors-20-02390-f012:**
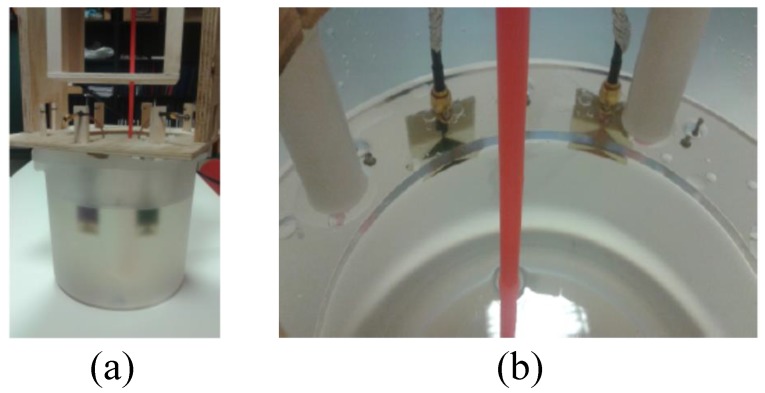
Photograph of a breast cancer detection system developed by Pagliari et al. [69]. (**a**) View of the antennas’ setup showing the breast holder consisting of a tank filled with a glycerine-water mixture. (**b**) Photograph of a vertically aligned cylinder representing the detection target. Reprinted with permission from [69].

**Figure 13 sensors-20-02390-f013:**
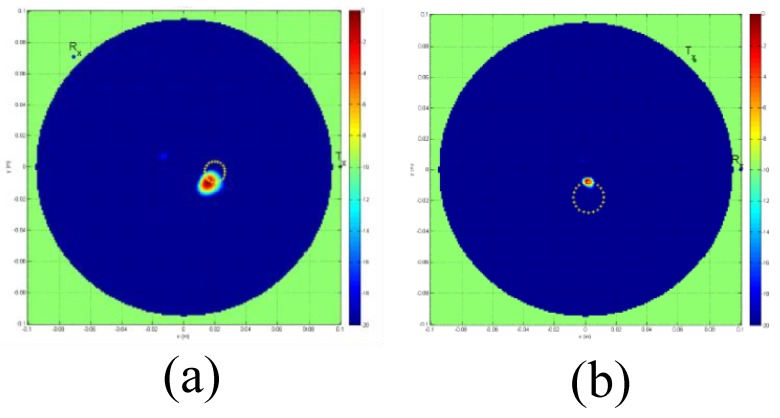
Reconstructed MT images obtained using the system developed by Pagliari et al. [69]. (**a**) A 12 mm tumour is indicated by the red spot inside a metallic cylinder breast phantom. (**b**) A 12 mm tumour is indicated by the red spot inside a dielectric cylinder breast phantom. Reprinted with permission from [69].

**Figure 14 sensors-20-02390-f014:**
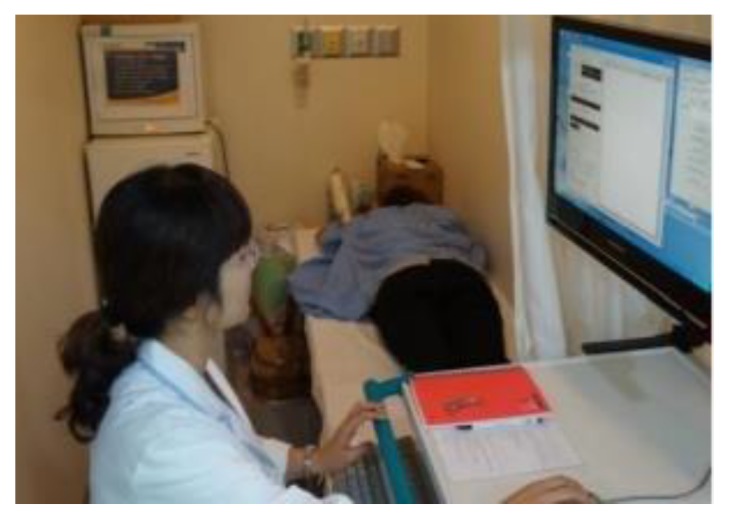
Clinical trial view of an MT test system developed by Jeon et al. [70]. Reprinted with permission from [70].

**Figure 15 sensors-20-02390-f015:**
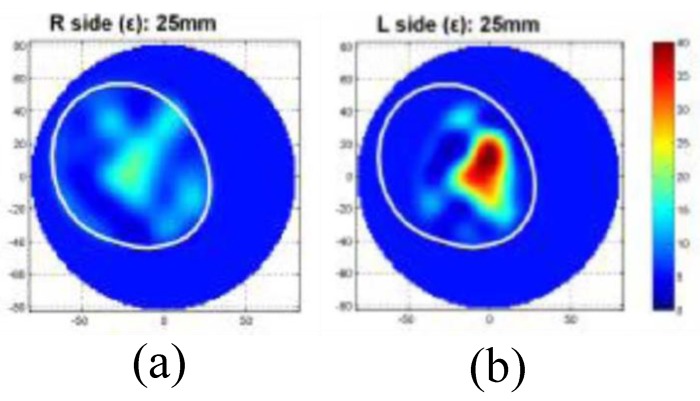
Reconstructed MT images obtained using the system developed by Jeon et al. [70]. (**a**) An image of the right tumour-free breast. (**b**) An image of the left breast, which indicates the presence of a 25 mm tumour. Reprinted with permission from [70].

**Figure 16 sensors-20-02390-f016:**
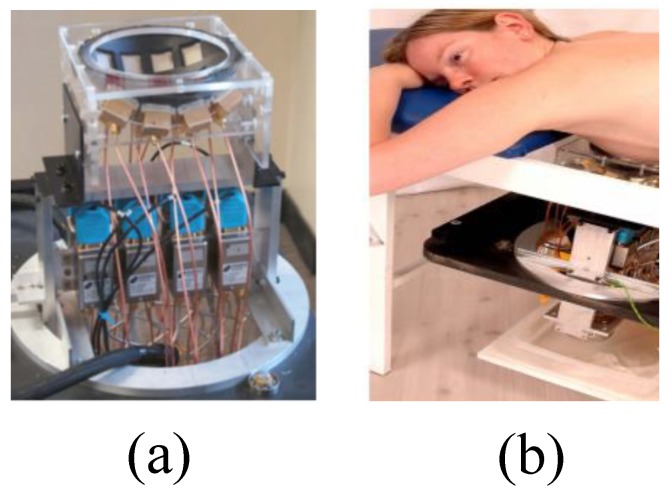
Prototype UWB frequency-domain radar-based MI system developed by Klemm et al. [76]. (**a**) View of the breast holder consisting of a conformal MI array, feed, and a switching system. (**b**) A photo of the prototype system setup for breast cancer detection showing a healthy female volunteer lying in a prone position. Reprinted with permission from [76].

**Figure 17 sensors-20-02390-f017:**
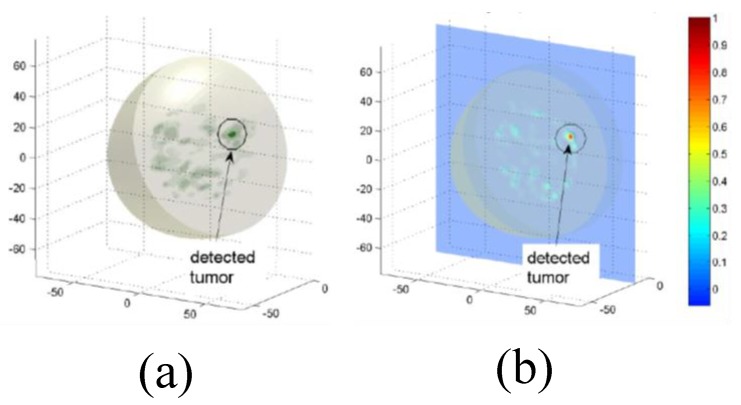
Clinical imaging results using a radar-based UWB microwave system developed by Klemm et al. [76]. (**a**) A 3D image. (**b**) A 2D image showing the presence of a tumour. Reprinted with permission from [76].

**Figure 18 sensors-20-02390-f018:**
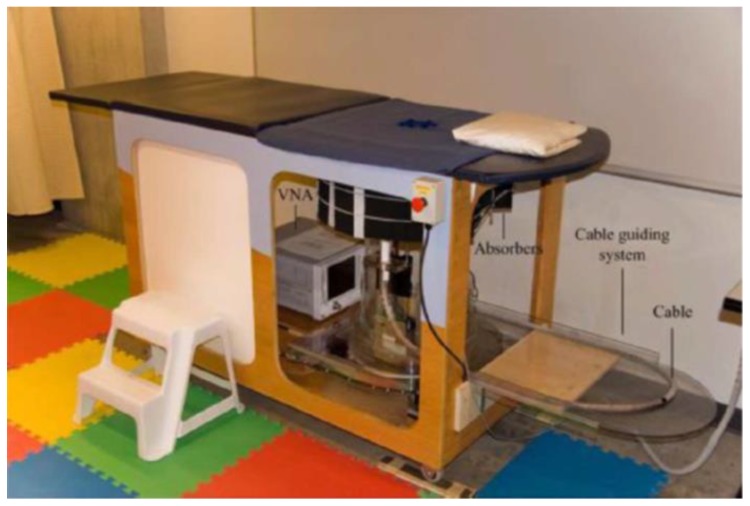
Tissue sensing adaptive radar (TSAR) prototype radar-based MI system for breast cancer detection developed by Fear et al. [78]. Reprinted with permission from [78].

**Figure 19 sensors-20-02390-f019:**
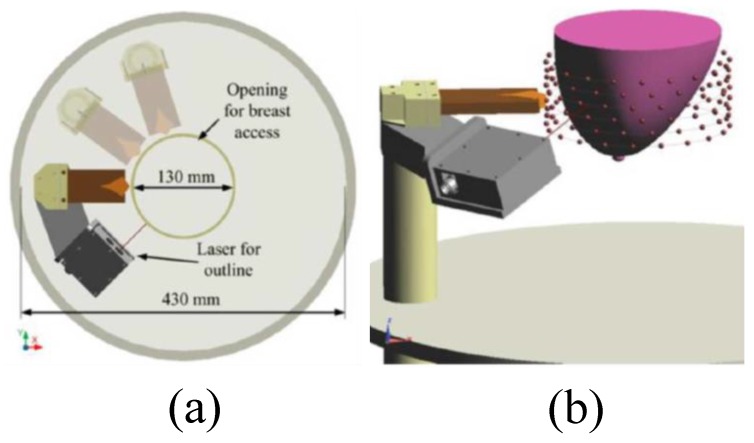
View of the radar-based MI system for breast cancer detection developed by Fear et al. [78]. (**a**) Top-down view of the antennas and laser setup. (**b**) Showing the breast holder consisting of a tank, laser, and antennas. Reprinted with permission from [78].

**Figure 20 sensors-20-02390-f020:**
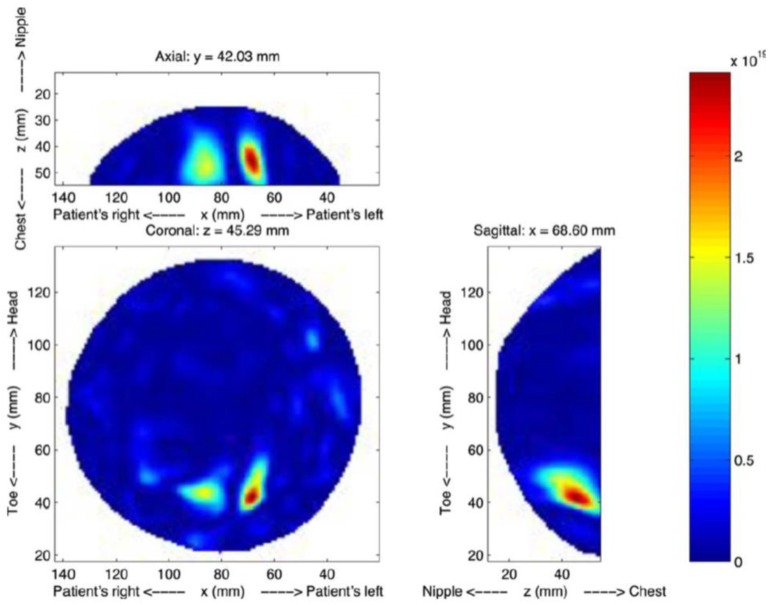
Reconstructed TSAR images using the system developed by Fear et al. [78]. Reprinted with permission from [78].

**Figure 21 sensors-20-02390-f021:**
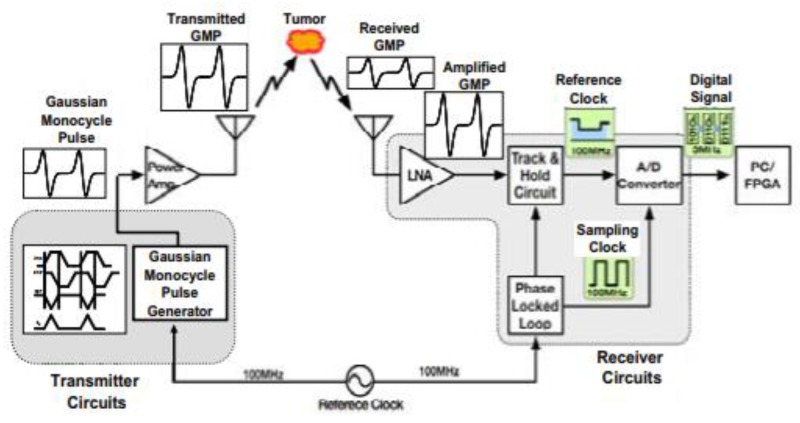
CMOS-based breast cancer detection system developed by Kikkawa et al. [79] consisting of transmitter and receiver circuits. The transmitter circuits contain a Gaussian mono-cycle pulse (GMP) generator, a power amplifier, and UWB antennas. The receiver circuits consist of a GMP receiver front-end low-noise amplifier (LNA) and a GMP data acquisition unit. Reprinted with permission from [79].

**Figure 22 sensors-20-02390-f022:**
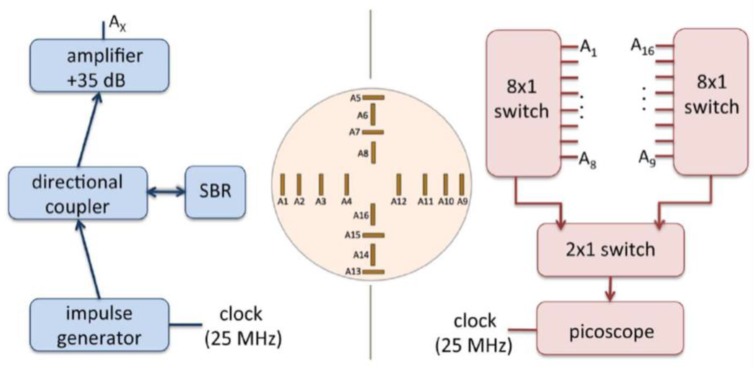
A schematic diagram representing the system developed by Porter et al. [80]. Reprinted with permission from [80].

**Figure 23 sensors-20-02390-f023:**
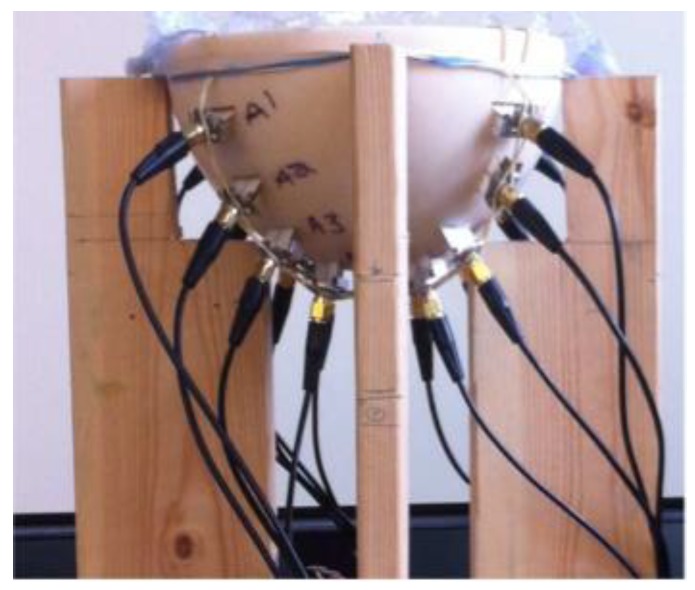
A breast cancer detection system using an antenna array with radome, developed by Porter et al. [80]. Reprinted with permission from [80].

**Figure 24 sensors-20-02390-f024:**
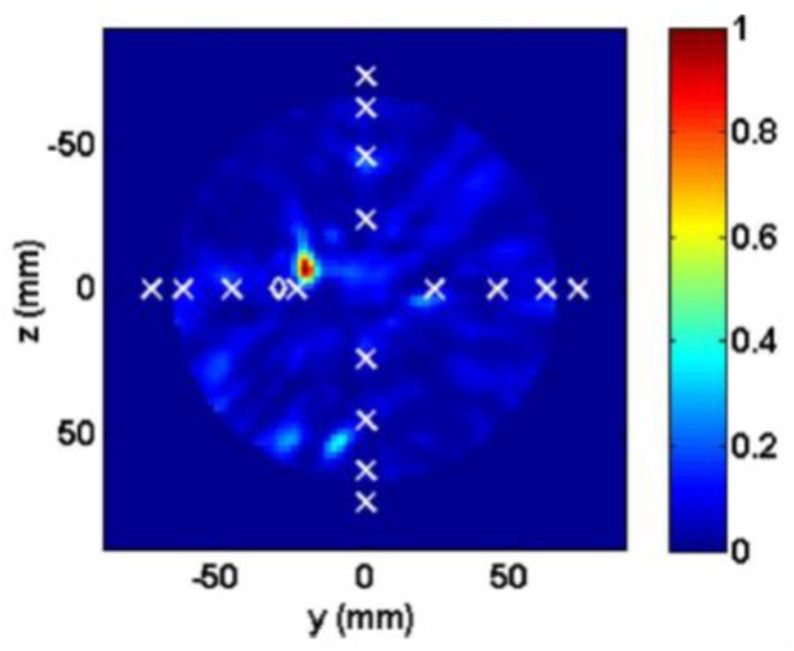
Reconstructed image of a breast phantom detection system developed by Porter et al. [80]. The tumour is indicated by the red spot; the x markers represent the positions of the antennas; and the diamond markers represent the actual location of the tumour centres. Reprinted with permission from [80].

**Figure 25 sensors-20-02390-f025:**
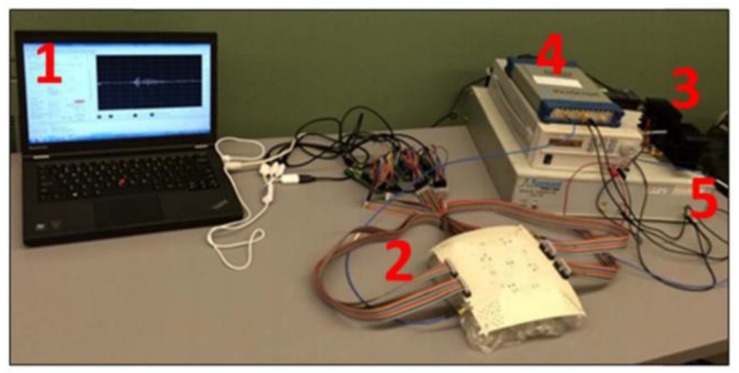
Microwave time-domain system for breast cancer imaging developed by Santorelli et al. [82]. The system components indicated on the photograph are (**1**) a laptop for transmitting control bits and saving the recorded data, (**2**) a flexible circuit board containing 16 antennas, the switching matrix, and the two micro-controllers, (**3**) the pulse shaping circuitry, (**4**) an oscilloscope, and (**5**) the impulse generator. Reprinted with permission from [82].

**Figure 26 sensors-20-02390-f026:**
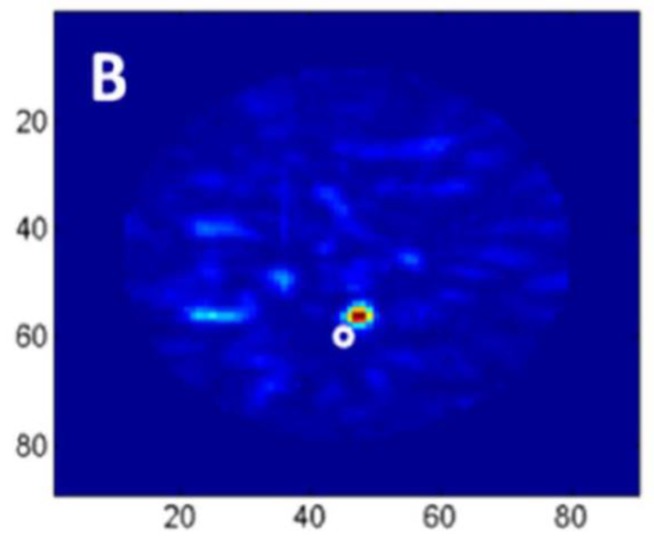
Reconstructed images of a rubber breast phantom obtained using the system developed by Santorelli et al. [82]. Reprinted with permission from [82].

**Figure 27 sensors-20-02390-f027:**
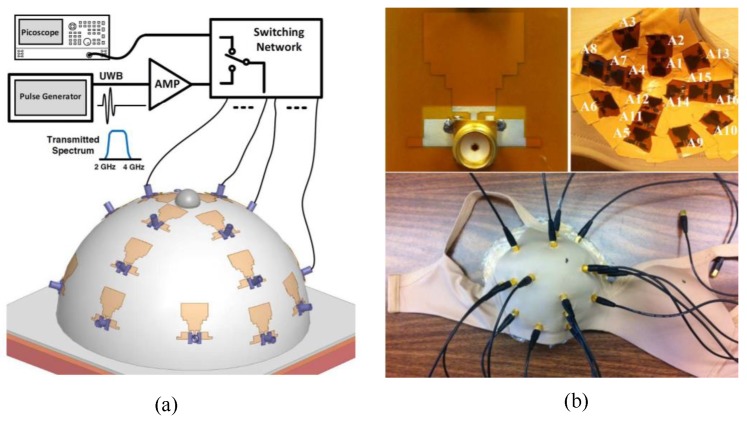
Microwave radar system for breast cancer detection developed by Porter et al. [83]. (**a**) A schematic of the microwave radar system setup. (**b**) A photograph showing the wearable prototype system. The top left photograph shows the connectorised monopole antenna. The top right photograph shows the antenna array inside the bra-cup with antennas’ numbers marked. The bottom photograph shows the outside of the bra and the cables that connect to the antennas. Reprinted with permission from [83].

**Figure 28 sensors-20-02390-f028:**
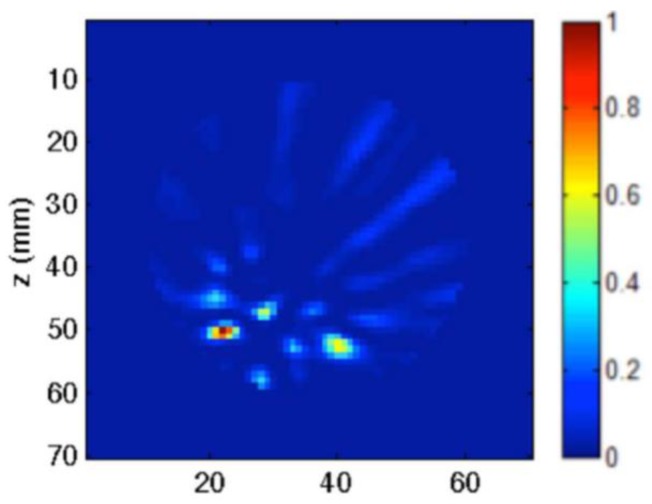
Reconstructed images for a breast slice taken at a depth of 6 mm from the chest wall obtained using the system developed by Porter et al. [83]. Reprinted with permission from [83].

**Table 1 sensors-20-02390-t001:** A comparison between microwave imaging (MI) and other breast cancer detection techniques.

Mammography	Ultrasound	MRI	Microwave Imaging
-Ionizing radiation	-Low sensitivity	-Very expensive	-Non-ionizing
-Uncomfortable: involves breast compression	-Higher cost than X-rays	-Some types of breast cancers cannot be detected	-Non-invasive -Inexpensive
-Sensitivity decreases with high density tissue	-A highly experienced operator is needed to perform an examination	-Inadequate positioning can cause an unsuccessful detection	-Comfortable

**Table 2 sensors-20-02390-t002:** Chronological summary of several studies of the dielectric properties of human female breast tissues.

Year	Author	Frequency Range	Probe Type	Tissue Type	No. of Samples (Patients)
1984	Chaudhary et al. [13]	3 MHz–3 GHz	Open-ended coaxial probe	Normal (unspecified) and malignant	15
1988	Surowiec et al. [15]	20 kHz–100 MHz	End-of-the-line capacitive probe	Normal (various) and malignant	7
1992	Campbell and Land [16]	3.2 GHz	Resonant cavity probe	Normal (various) and benign, malignant	37
1994	Joines et al. [17]	50–900 MHz	Flat-ended coaxial probe	Normal (unspecified) and malignant	12
2004	Choi et al. [18]	0.5–30 GHz	Open-ended coaxial probe	Lymph nodes and malignant	12
2007	Lazebnik et al. [19]	0.5–20 GHz	Open-ended coaxial probe	Normal (adipose and glandular) and malignant	196
2009	Halter et al. [20]	100 Hz–8.5 GHz	Electrical impedance spectroscopy probe	Fibrocystic and adipose and tumour tissues	6
2014	Sugitani et al. [21]	1–6 GHz	Open-ended coaxial probe	Fibroglandular and tumour tissues	102
2015	Martellosio et al. [22]	0.5–50 GHz	Open-ended coaxial probe	Adipose and tumour tissues	Unspecified
2017	Martellosio et al. [23]	0.5–50 GHz	Open-ended coaxial probe	Adipose and tumour tissues	220
2018	Cheng et al. [28]	0.5–8 GHz	Open-ended coaxial probe	Normal (unspecified) benign and tumour	509
2019	Hussein et al. [29]	200 MHz to 13.6 GHz	Open-ended coaxial probe	Normal (unspecified) and tumour tissues	48

**Table 3 sensors-20-02390-t003:** Summary of some clinical MI studies.

Research Group	Scan Time	Max No. of Patients	Coupling Medium	Examination Table	Antenna Array Type	Acquisition and Imaging	Position
Dartmouth College [44,102,103,104,105,106]	5 min	175	✓	✓	Monopole synthetic	Frequency, tomography	Prone
		medium				
University of Calgary [78,107,108,109]	30 min	8	✓	✓	Vivaldi synthetic	Frequency, DAS	Prone
		medium				
McGill University [110]	5 min	13	✓	✗	Microstrip stationary	Time, DAS	Seated
		shell				
Bristol University [111,112,113,114]	10 S	267	✓	✓	Slot Hardware	Frequency, IDAS	Prone
		shell				
Shizuoka University [115,116]	3 min	2	✓	✓	Stacked patch hardware	Frequency, DAS	Prone
		shell				
Southern University of Science and Technology [117]	4 min	11	✓	✓	Horn synthetic	Frequency, DAS	Prone
		medium				
Hiroshima University [118]	5 min	14	✓	✗	Planar slot synthetic	Time, DAS	Supine
		shell				
London South Bank University [119]	5 min	45	✓	✓	Planar antenna synthetic	Frequency, neural network	Prone
		shell

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
