# Peer review of "Review of Microwaves Techniques for Breast Cancer Detection"

_sensors, 2020, doi:10.3390/s20082390_

Round 1

Reviewer 1 Report

The authors attempt to "encourage a healthy debate on why a commercially available system has yet to be made available despite almost thirty years of intensive research". In my opinion, they do not achieve this goal.

A number of specific issues:

1) MARIA® is arguable commercially available as they have a distributor. Two systems are being developed by industrial researchers.

2) lines 32–36: this description is for asymptomatic screening. Ultrasound is indicated for distinguishing solid and cystic masses discovered on mammogram for example.

3) lines 46:52: given the amount of reviews of microwave breast imaging (almost too many to list: books by Nikolova and Conçeicão, reviews by Moloney et al., O'Loughlin et al., Modiri et al. etc. etc. etc.).

4) The inclusion criteria for the systems are not clear. Why are these systems being reviewed and what can be concluded from that?

5) The divisions in section 3 are not sufficient, there are no technical barriers to using the same collected signals in a radar-based and tomographic system in many cases and they are presented as completely separate approaches.

6) Lines 243–247: These are not different approaches, nor are these all imaging algorithms.

In general, I do not feel the conclusions support the results at all.

Reviewer 2 Report

This is a very comprehensive review on microwaves techniques for breast cancer detection. However, there are some points need to be improved.

  1. For better attracting the audience who are not in microwave imaging field, some introduction of background is necessary before talking about microwave dielectric properties of breast tissues.
  2. What is the advantage of microwaves techniques for breast cancer detection compared with other detection methods? Some detailed comparison and analysis are suggested to be added.

Reviewer 3 Report

Please find my review in the attached file.

Round 2

Reviewer 1 Report

I'd like to thank the authors for their response. I don't agree with some of the responses as listed below.

Previous comments:
2) "However, ultrasound is not advantageous for breast imaging as it exhibits low resolution for not being able to distinguish between malignant and benign breast tumors [7] -[8]"
As I said previously, this is quite misleading. Ultrasound breast screening was not found to be useful, although similar could be said about mammography breast screening. Ultrasound is the primary way of distinguishing between solid and cystic masses detected on a mammogram, so to say it is not advantageous is quite misleading.

3) In my opinion, a chronological summary is not a review.

5) I believe Section 3 is highly misleading. There is a lot of value in comparing the hardware of systems designed with radar-based imaging in mind compared to those of tomography in mind, however, this is entirely unclear from this section. The classification does not, in any way, apply to the hardware, it applies to the reconstruction algorithms. This is not at all clear.

6) CM and TSAR can not conceivably be described as different approaches. The response does not address this fundamental problem.

Other comments:

Table 3 is remarkably similar to that in O'Loughlin et al.

Conclusions:

No evidence is presented that a matching liquid leads to a lower signal to noise ratio.

It is not clear to me why a "chronological summary" is useful in identifying why there are few commercial systems available.

Reviewer 3 Report

The authors have fixed all my concerns.